# PIEZO1 loss-of-function compound heterozygous mutations in the rare congenital human disorder Prune Belly Syndrome

Nathalia G. Amado[1,4,5], Elena D. Nosyreva[2,5], David Thompson[2], Thomas J. Egeland[1], Osita W. Ogujiofor[2], Michelle Yang[1], Alexandria N. Fusco[1], Niccolo Passoni[1], Jeremy Mathews[3], Brandi Cantarel[3], Linda A. Baker ®[1,4] ✉ & Ruhma Syeda ®[2] ✉

Prune belly syndrome (PBS), also known as Eagle-Barret syndrome, is a rare, multi-system congenital myopathy primarily affecting males. Phenotypically, PBS cases manifest three cardinal pathological features: urinary tract dilation with poorly contractile smooth muscle, wrinkled flaccid ventral abdominal wall with skeletal muscle deficiency, and intra-abdominal undescended testes. Genetically, PBS is poorly understood. After performing whole exome sequencing in PBS patients, we identify one compound heterozygous variant in the PIEZO1 gene. PIEZO1 is a cation-selective channel activated by various mechanical forces and widely expressed throughout the lower urinary tract. Here we conduct an extensive functional analysis of the PIEZO1 PBS variants that reveal loss-of-function characteristics in the pressure-induced normalized open probability (NPo) of the channel, while no change is observed in single-channel currents. Furthermore, Yoda1, a PIEZO1 activator, can rescue the NPo defect of the PBS mutant channels. Thus, PIEZO1 mutations may be causal for PBS and the in vitro cellular pathophysiological phenotype could be rescued by the small molecule, Yoda1. Activation of PIEZO1 might provide a promising means of treating PBS and other related bladder dysfunctional states.

Prune Belly Syndrome (PBS), also known as Eagle-Barrett or Triad Syndrome (OMIM#100100) is a rare, multi-system congenital myopathy primarily affecting males and incompletely understood genetically. Phenotypically, its morbidity spans from mild to lethal. The classic phenotypic triad can vary in severity but includes (1) 'prune-like' wrinkled, flaccid ventral abdominal wall with regionally absent or hypoplastic skeletal muscle, (2) urinary tract dilation including megabladder and megaureter with poorly contractile smooth muscle, and (3) bilateral intraabdominal cryptorchidism[1–3]. The ability to sense and respond to mechanical force is critical for the proper function of the urinary bladder[4,5]. The organ experiences a range of direct pressures on its smooth muscle wall and urothelial cell lining during urine storage and elimination. The combined organ phenotypes observed and the under-active bladder in PBS patients suggest that, at the

[1]Department of Urology, University of Texas Southwestern Medical Center, Dallas, TX, USA. [2]Department of Neuroscience, University of Texas Southwestern Medical Center, Dallas, TX, USA. [3]Department of Bioinformatics, University of Texas Southwestern Medical Center, Dallas, TX, USA. [4]Present address: The Kidney and Urinary Tract Center, The Abigail Wexner Research Institute at Nationwide Children's Hospital, Columbus, OH, USA. [5]These authors contributed equally: Nathalia G. Amado, Elena D. Nosyreva. ✉e-mail: Linda.Baker1@nationwidechildrens.org; Ruhma.Syeda@utsouthwestern.edu

cellular level, there is an alteration in mechanosensation. However, the cause of most PBS cases has not been fully elucidated. In this study, we identify one PBS-affected individual carrying compound heterozygous mutations in the PIEZO1 gene.

Discovered by Patapoutian and colleagues[6], the evolutionarily conserved PIEZO protein family are cell membrane-embedded mechano-transducing ion channels that convert cellular physical forces (shear and tensile or compressive stress) into electro-chemical responses[7–9]. The PIEZO1 gene encodes for a protein monomer of the mature propeller-shaped trimeric ion-channel. The PIEZO1 trimer senses forces directly imparted by lipid membranes through its membrane-embedded blades, opens its membrane-spanning central pore for cation ($Ca^{++}$, $K^+$, $Na^+$) influx thereby generating cellular current, and governs the probability of its open versus closed state[8–13]. PIEZO1 is widely expressed in non-neuronal cells and is a critical sensor of red blood cell volume regulation and shear stress[14–17]. Moreover, PIEZO1 is highly expressed in the human and murine bladder urothelium and detrusor, as well as in vascular smooth muscle cells (SMCs)[18–20]. In mammals, PIEZO1 is required for regulation of urinary osmolarity[21], proper vascular development and cell rearrangements in response to flow[22,23], stretch-triggered epithelial proliferation[24], and neuronal stem cell lineage choice[25]. So far, PIEZO1 mutations have not been associated with any human congenital urinary tract disorders[20]. Here, we synergize basic science and clinical efforts to characterize the clinical phenotype, the genotype, and the cellular pathophysiology of the detected rare compound heterozygous PIEZO1 missense mutations.

## Results

### Clinical phenotype of the PBS proband carrying compound heterozygous mutations in the PIEZO1 gene

This proband is a white non-Hispanic adult male born full-term via spontaneous vaginal delivery to a mother with no smoking history but with gestational exposure to father's smoking. Otherwise, there was no family history of PBS (Tables 1 and 2, Fig. 1a).

At birth, the proband's physical exam was consistent with PBS, including abdominal wall laxity, flared ribs, and dimpled knees. Imaging identified dilated bladder and bilateral hydroureter. During his 3-week NICU stay, he required surgical urinary bladder drainage by a bladder stoma (vesicostomy). At one year of age, surgery for his undescended testicles was performed (bilateral orchiopexy). Additionally, he underwent bilateral tapered ureteral reimplantation at age 2 years for moderate vesicoureteral reflux and recurrent urinary tract infections. At 7 years old, a major abdominoplasty was performed to tighten the lax abdominal wall, the vesicostomy was surgically closed, and the appendix was detached from the intestine and reconfigured to span from the urinary bladder to the umbilical skin to permit daily intermittent passage of a catheter for urinary bladder emptying (a

Mitrofanoff procedure) due to ongoing inability to completely empty his bladder. Developmentally, no intellectual or other motor delay was observed.

As an adult, he has bilateral hydronephrosis with left and right kidney function at 30% and 70%, respectively and stage 2 chronic kidney disease with proteinuria. He has not used a urinary catheter in his Mitrofanoff in the past 20 years and is voiding per urethra. Of note, he has undergone a cardiac ablation to treat erratic episodes of supraventricular tachycardia. He had pneumonia after surgery but overall, he reports no issues with his lungs. Additionally, he routinely manages constipation through diet and laxatives as needed. His case reveals that PBS is a multi-organ system congenital disorder that can be individually variable in severity.

To calculate his clinical PBS phenotype disease severity, we employed a scoring system that our research group developed called the RUBACE score (R: renal, U: ureter, B: bladder/outlet, A: abdominal wall, C: cryptorchidism, E: extra-genitourinary)[26]. The proband's RUBACE score was 15, indicating moderately severe PBS based on the following sub-scores: (R)enal GFR > 60 (1 point), (U)reteral: reimplanted ureters, hydroureter, tapered ureters, moderate reflux (2 points), (B)ladder/Outlet: Mitrofanoff/appendicovesicostomy (3 points), (A)bdominal Wall: abdominoplasty (2 points), (C)ryptorchidism: orchiopexy (2 points), (E)xtra-genitourinary Manifestations: Neurologic (0), Cardiac: cardiac ablation (2 points), GI: laxatives (1 point point), MSK: flared ribs, dimpled knees (1), and Respiratory: Pneumonia (1 point). His medical, surgical, and family history is summarized in Tables 1 and 2.

### WES identifies mutations in PIEZO1 in PBS proband

Peripheral blood leukocyte DNA was extracted from the affected PBS subject for paired end whole exome sequencing (WES). Details on WES variant filtering and analysis are provided in Supplementary Table 1. Exceedingly rare, potentially impactful variants that met our filtering criteria (Table 3) included two heterozygous PIEZO1 variants (Genebank NM001142864.4) - c.757 G > A;p.Gly253Arg (murine Ser260Arg) and c.6584 C > T;p.Ser2195Leu (murine Ser2211Leu) on human chromosome 16. Sanger sequencing of the proband and his parents confirmed that the PBS proband is compound heterozygous, having inherited the c.757 G > A - p.Gly253Arg variant from his carrier asymptomatic mother and the c.6584 C > T - p.Ser2195Leu variant from his carrier asymptomatic father (Fig. 1a and Supplementary Fig. 1). No DNA was available from the unaffected brother and the spontaneous aborted fetus.

The full-length human PIEZO1 (human PIEZO1) has 2521 amino acids. The cryo-electron microscopy (cryo-EM) structure of mouse Piezo1 (mPiezo1) reveals that it trimerizes to form a unique three-bladed, propeller-shaped architecture, comprising a central ion-conducting pore module and three peripheral blades[12,27–29] (Fig. 1b, c). The central pore is encoded by the C-terminal residues and the rest of the protein forms the peripheral blades and beam (Fig. 1c)[30]. The p.S2195L mutation within PIEZO1 affects a highly conserved residue on the transmembrane domain 37 (the outer helix) of the central pore domain (Fig. 1b–d). The G253R mutation is localized on the transmembrane domain 7, a region that is not been structurally resolved yet. The sequence alignment (Fig. 1c, d) showed that mouse Piezo1 harbors a Serine (S), instead of a Glycine (G) found in human PIEZO1 at this position. By performing the same biophysical tests described below, we confirmed that there are no functional differences in S to G substitution at human aa253 as measured by single channel conductance and channel normalized open probability (NPo) acquired at −60 mV (Supplementary Fig. 2). Considering together the exceedingly rare allelic frequency, the in silico predicted damaging impact of the amino acid changes (Table1), and the evolutionary conservation, this evidence warranted investigation for pathophysiological impact of the compound heterozygous PIEZO1 variants in the PBS proband.

## Table 1 | PBS subject's medical and family history

| | General Information | Smoking | Drinking |
|---|---|---|---|
| PBS Subject | See details below* | Unknown | Unknown |
| Mother | 1 miscarriage (lost at 1 month) | NO | NO |
| Father | Acid Reflux | YES | Unknown |
| Brother | Aortic Aneurysm | Unknown | Unknown |
| MGM | Emphysema | YES | YES |
| MGF | Died of heart attack | Unknown | Unknown |
| PGF | Unknown | YES | Unknown |

The PBS subject's family history was limited. No information was available for paternal grandmother.
*MGF* maternal grandfather, *MGM* maternal grandmother, *PGF* paternal grandfather.

**Table 2 | PBS Subject's Detailed Medical and Surgical History**

| Bladder | Urinates through urethra, APV, childhood bladder infections |
|---|---|
| Ureter | Hydroureter, moderate reflux. |
| Kidney | Hydronephrosis bilaterally with 30% & 70% renal function, CKD grade 2 |
| GI | Constipation managed by diet and laxatives |
| Heart | Random episodes of SVT and catheter ablation at teenager |
| MSK | Underdeveloped abdominal muscles, flared rib, and dimpled knees |
| Surgical history | Vesicostomy, mitrofanoff, abdominoplasty, bilateral orchiopexy, and heart surgery |
| PBS Severity (RUBACE score) | RUBACE 15 (moderate) |

RUBACE scoring system based on six sub-scores (R: renal, U: ureter, B: bladder/outlet, A: abdominal wall, C: cryptorchidism, E: extra-genitourinary, generating the acronym RUBACE), yielding a potential summed score of 0–31.

*CKD* chronic kidney disease, *APV* appendicovesicostomy, *GI* gastrointestinal, *SVT* Supraventricular tachycardia, *MSK* musculoskeletal.

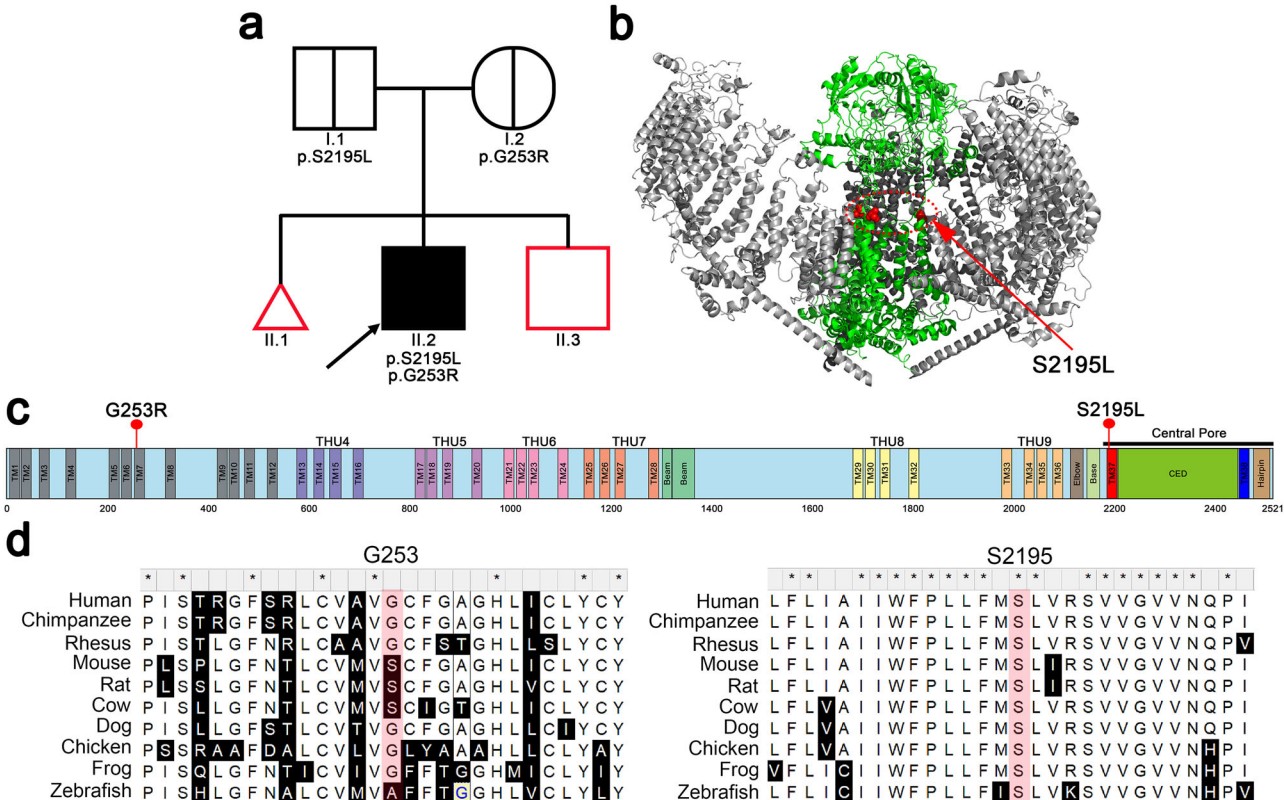

**Fig. 1 | Pedigree and *PIEZO1* compound heterozygous mutations in a PBS patient. a** Pedigree of the proband family showing the origin of his compound heterozygous *PIEZO1* variants. The affected male (II.2) is indicated with an arrow and a black square. The carrier father (I.1) and the carrier mother (I.2) are shown by a central black line. Individuals without available DNA are outlined in red (II.1 and II.3). Males = square, female = circle, miscarriage unknown sex = triangle. **b** Cryo-EM trimeric mPiezo1 structure (Protein Data Bank ID #6B3R). mPiezo1 blades (gray) encompass residues 577–2123. Pore domain (green) spans from 2124 to 2547 residues. Human PBS pore domain variant at S2195L (murine S2211L) is shown as red

spheres on all three monomers. Since amino acids 1-576 are not structurally resolved, the human PBS variant G253R (murine S260R) is not shown but is predicted to be within the lipid bilayer. **c** Schematic representation of the missense *PIEZO1* variants in PBS male in the region encoding the protein domains. **d** Multiple sequence alignment on *PIEZO1* mutated residues, showing that the G253 residue is not conserved in mouse, rat, cow, and zebrafish. However, the S2195 mutated residue is highly conserved among vertebrate species analyzed, suggesting that this residue plays an important role in PIEZO1 function.

## Biophysical properties of PIEZO1 PBS Variants

Given PIEZO1 is a mechanosensitive channel, we performed cell-attached electrophysiology recordings while applying direct pressure to the cell membrane, to mirror the mechanical stimuli experienced by the urinary bladder. HEK293T$^{\Delta P1}$ cells are null for PIEZO1. mPiezo1 or PBS compound heterozygous variants S260R and S2211L were expressed in HEK293T$^{\Delta P1}$ cells to obtain single channel permeation (current) and gating (NPo) properties.

First, wild type (WT) mPiezo1 exhibits single channel currents = $-1.45 \pm 0.1$ pA at $-60$ mV after application of $-30$ mmHg pressure in

agreement with the previously reported values[30,31]. No significant differences were found in single channel current properties of PBS variants either expressed alone S260R = $-1.42 \pm 0.16$ pA, S2211L = $-1.4 \pm 0.13$ pA or when the mutants were co-expressed S260R + S2211L = $-1.4 \pm 0.08$ (Fig. 2a, b). This is reflective in the single channel conductance values of the WT and mutant channels, which are not significantly different when assayed at $-60$ mV (Fig. 2d).

Second, we assessed the gating function by measuring the NPo of the transfected WT and PBS mutant channels before and after application of $-30$ mmHg pressure. Single transfections of WT and the PBS

## Table 3 | PBS subject's PIEZO1 variants' information

| Amino Acid Variant (*Homo Sapiens*) | p.S2195L | p.G253R |
|---|---|---|
| Amino Acid Variant (*Mus musculus*) | S2211L | S260R |
| Allele Frequency General Population | 0.00009197 | 0.0001708 |
| Allele Frequency Caucasian Male | 0.00001271 | 0.00006974 |
| VEST | 0.911 | 0.719 |
| SIFT | Deleterious (0.002) | Deleterious (0.014) |
| PP2HVAR | Damaging (0.989) | Probably Damaging (0.546) |
| PP2HDIV | Damaging (1.0) | Probably Damaging (0.931) |
| Mutation Taster | Disease causing | Polymorphism |
| Mutation Assessor | Medium | Neutral |
| LRT | Deleterious (0.00000) | Not available |
| GERP | 4.68 | 2.34 |
| CAAD | 57 | 23.4 |

Piezo1 variant general information and protein function prediction. VEST (Variant effect scoring tool) assigns variants a score between 0 and 1, where 1 indicates a confident prediction of a functional mutation. SIFT (Sorting Intolerant from Tolerant) score ranges from 0.0 to 1. The score between 0.0 and 0.05 are considered deleterious. The PP2HVAR (PolyPhen2 All Human Variants) and PP2HDIV (PolyPhen2 Disease Human Variants) score ranges from 0.0 to 1.0. The score between 0.0 and 0.15 are predicted to be benign, 0.16 to 0.84 are possibly damaging, and 0.85–1.0 are more confidently predicted to be damaging. MutationTaster score is the probability that the prediction is true. Mutation Taster predicts an alteration as one of four possible types: DC - disease causing (probably deleterious), A - disease causing automatic (known to be deleterious), N - polymorphism (probably harmless), or P - polymorphism automatic (known to be harmless). Mutation assessor uses a multiple sequence alignment to reflect functional specificity and generates conservation scores to represent the functional impact of a missense variant. Variants classed as neutral (NE) or low (L) are predicted to not impact protein function, whereas variants classed as medium (M) or high (H) are predicted to result in altered function. LRT (likehood ratio test) detected the significant rate shifts on acid conservation at specific sites in proteins. The score variant can range from 0 to 1, with lower scores reflecting greater likelihood that the variant is disease-causing. GERP (Genomic Evolutionary Rate Profiling) score is a measure of sequence conservation across multiple species. It ranges from minus 12.3 (–12.3) to 6.17, with 6.17 being the most conserved. CAAD (Combined Annotation Dependent Depletion) is a framework that integrates multiple annotations into one metric by contrasting variants that survived natural selection with simulated mutations. The score ranges from 1 to 99, higher values indicating more deleterious cases which were classified as benign up to 25, and anything above was classified as pathogenic.

mutants respond with a significant increase in NPo when comparing before and after pressure application (Fig. 2c). Specifically, NPo increased after pressure application in WT mPiezo1, S260R expressed alone, and S2211L expressed alone by 4, 1.7, and 3-fold, respectively (Fig. 2c). In contrast, when the mutants were co-expressed, no significant change in NPo was observed before versus after pressure application; S260R + S2211L = 1.1-fold change (Fig. 2c).

Third, when comparing the open probability of channels obtained from four distinct transfection conditions to each other after application of −30 mmHg pressure, the pore domain mutant S2211L exhibit NPo = 0.07 ± 0.04, and co-transfection of S2211L + S260R has NPo = 0.09 ± 0.05 which are the lowest NPo values. The PBS variant either expressed alone or co-transfected showed significantly lower NPo values than WT NPo = 0.26 ± 0.07 when statistically assessed using one-way ANOVA test (Fig. 2e). Additionally single channel open probability was also assayed in response to a range of pressures (from −10 to −50 mmHg). In line with the data presented at −30 mmHg, both the mutants S260R and S2211L exhibit lower NPo than WT Piezo1 at various applied pressure, strengthening the point that the mutants have lower sensitivity to pressure even at different pressures (Supplementary Fig. 3).

Furthermore, to complement the open probability data, we performed dwell time analysis to calculate the mean open time of the channel. The mean open times of WT Piezo1, S260R, S2211L and co-expressed channels were analyzed by constructing dwell time histograms (Fig. 2f and Supplementary Fig. 4). In line with the NPo data, the mean open time of mutants were significantly lower than WT suggesting that shorter open dwell times drives the decreased NPo in the mutants. Combined, this clearly shows that PBS point mutations have intact single channel conductance at −60 mV but impaired pressure sensing and gating properties when expressed alone or when the mutants were co-expressed.

### PIEZO1 PBS mutants are loss-of-function

To assess for a trafficking and functional effect, we co-expressed WT mPiezo1 fused with tdTom and the PBS mutants S260R and S2211L fused with GFP. Confocal studies showed that WT-tdTom and mutant-GFP channels are co-expressed in HEK293T$^{\Delta P1}$ cells as seen by the respective red and green signal (Fig. 3a). Furthermore, both mutants, either expressed alone or co-expressed, are trafficked to the plasma membrane surface, when assayed qualitatively with the extracellular myc-tag labeling (Supplementary Fig. 5). Further quantitative experiments with thousands of cells, such as flow cytometry, could be used to assess the exact percentage of mutant's surface expression in comparison to WT Piezo1, since our qualitative analysis imaged 9 cells per condition. For electrophysiology experiments, only those cells were patched where both red and green signals were visible to ensure that cells have both the WT mPiezo1 (red) and PBS mutants (green). To test that mPiezo1 single-channel properties are not altered by the fused tags, we compared WT-tdTom, WT-GFP and WT without any fused tag. No significant change in single-channel conductance or NPo (measured at −60 mV) was observed in three different WT-mPiezo1 suggesting that addition of tdTom or GFP does not alter protein channel function (Supplementary Fig. 6). Next, WT-tdTom was co-expressed with either S260R-GFP or S2211L-GFP. In line with our previous data, no significant differences were found in single-channel current of PBS variants when expressed with WT-tdTom (Fig. 3b, c). However, significant decreases in both mutant's NPo were observed when co-expressed with WT mPiezo1 NPo = 0.26 ± 0.06, S2211L NPo = 0.07 ± 0.03 and S260R NPo = 0.13 ± 0.06 (Fig. 3d). Together, these in vitro experiments confirm that PBS mutants S260R and S2211L independently act in a loss-of-function fashion.

### Yoda1 positively modulates PBS mPiezo1 channel activity

Yoda1 is a small molecule agonist activating both mouse and human PIEZO1[14,32]. To assess for drug targeting possibilities of mutant channels, we tested whether Yoda1 can fully or partially rescue the function of PBS PIEZO1 variants. We performed cell-attached electrophysiology on HEK293T$^{\Delta P1}$ cells either expressing WT mPiezo1, S260R, S2211L or co-expressed S260R + S2211L (Fig. 4). WT mPiezo1 responded to 20 µM Yoda1 exhibiting 1.9-fold increase in NPo. Surprisingly, both mutants S260R and S2211L, either expressed alone or co-expressed, showed a Yoda1-dependent increase of 3.2, 3.4, and 2.3-fold in NPo (Fig. 4a, c). Additionally, no change in the single channel currents were observed when the all-point current histograms were overlayed with and without Yoda1, suggesting that Yoda1 only changes the gating properties of the mutant channels (Fig. 4b). No significant differences were found in the NPo when WT Piezo1 without Yoda1 treatment was compared with S2211L expressed alone or co-expressed with S260R in the presence of Yoda1, suggesting that Yoda1 rescued the open probability of faulty mutants to near WT levels.

In addition to single-channel electrophysiology, we evaluated whole cell calcium influx via cell-based fluorescence assay using Fluo-4 dye in HEK293T$^{\Delta P1}$ cells transfected with either WT mPiezo1 or PBS mutants. Correlating with our previous data, Yoda1 affected intracellular calcium influx on WT and PBS mPiezo1 mutations, suggesting that channels are chemically stimulated in the absence of applied mechanical forces (such as pressure) (Fig. 5a). Neither the DMSO

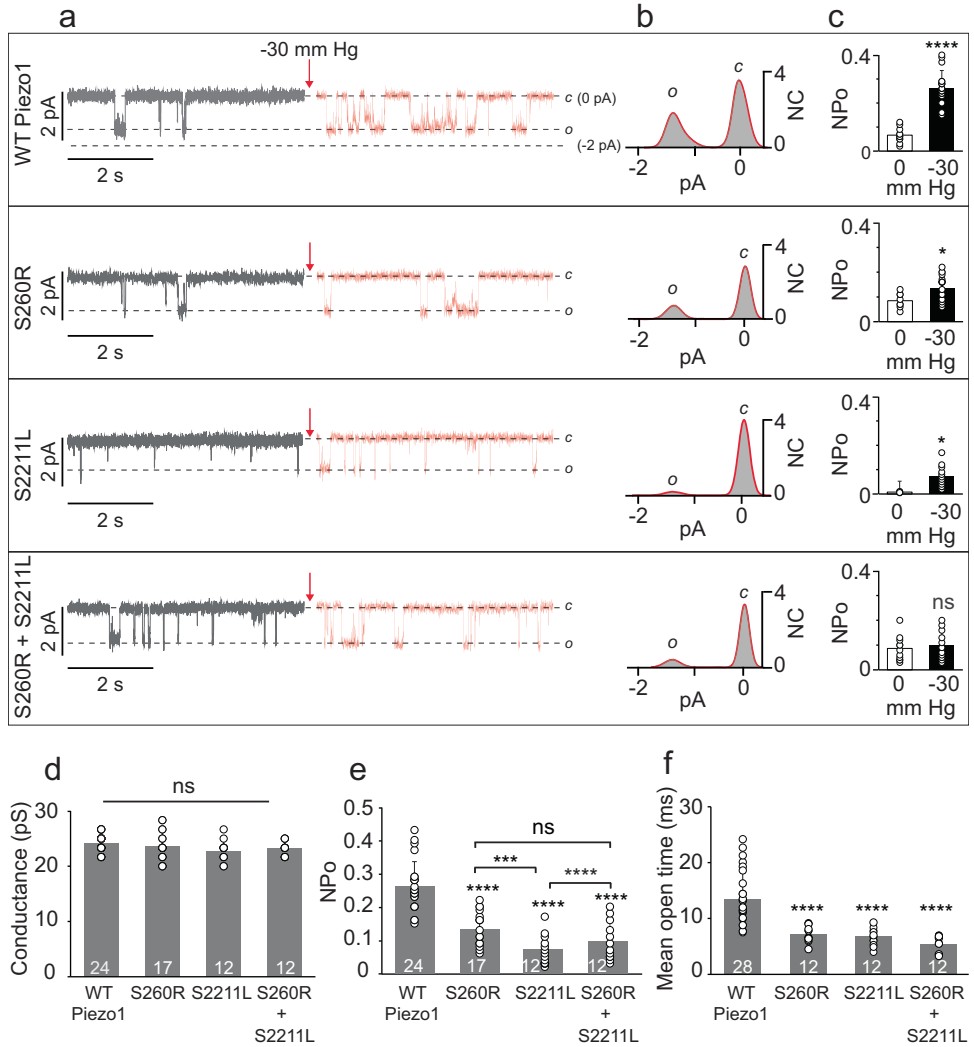

**Fig. 2 | Single-channel Properties of WT versus PBS Piezo1 Variants.** HEK293T$^{\Delta P1}$ cell attached patch clamp experiments were performed without and with applied pressure (−30 mmHg) at −60 mV. **a** Single-channel current recordings of WT and PBS variant proteins S260R, S2211L and co-expressed S260R + S2211L, showing the closed (C) - 0 pA and open (O) -−1.5 pA state of the channel before (gray) and after (red) pressure application (red arrow). The pipette solution contained 130 mM Na$^+$ and 1 mM Ca$^{++}$, the bath solution contained 140 mM K$^+$ and 1 mM Mg$^{++}$. **b** All-point current histograms of the single-channel recordings shown in (**a**) (red) acquired under −30 mmHg pressure. The area under the curve demonstrates the C and O states of the channel. X-axis is the current amplitude in picoamperes (pA). Y-axis is Normalized Counts (NC). **c** Steady state NPo of WT and PBS variants either without

any applied pressure to the patch of membrane (white bars) or with −30 mmHg applied pressure (black bars). X-axis is pressure in mmHg. Y-axis is NPo. (WT $n = 24$, S260R $n = 17$, S2211L $n = 12$, S260R + S2211L $n = 12$). **d** Single channel conductance of WT and PBS variant channels obtained from panel B at −60 mV. Experimental replica numbers are shown in the bar graph ($n > 12$). **e** Steady state NPo at −30 mmHg where experimental replica numbers are shown in the bar graph ($n > 12$) and (**f**) mean open time of WT and PBS variants ($n > 12$). Statistical analysis was performed by using Student's t test (**c**) and One-way ANOVA (**d–f**) where *$p < 0.05$, **$p < 0.01$, ***$p < 0.001$, ****$p < 0.0001$ and ns = not significant. The data in bar graphs in (**c–f**) are presented as Mean ± SD.

control or the Mock transfection induced calcium influx in any of the tested conditions (Fig. 5a, b). Total cell fluorescence of WT mPiezo1 and S260R are not statistically different, while S2211L and co-expressed S260R + S2211L achieve 66% and 72% of WT channel activity, respectively (Fig. 5c). In conclusion, using two functional assays, we show that the PIEZO1 PBS variants' gating properties can be restored by Yoda1.

## Discussion

In this study, we expanded the phenotypic spectrum of PIEZO1-related diseases by identifying by WES a PBS male individual who is compound heterozygous for rare missense mutations in PIEZO1 that lead to PIEZO1 protein functional compromise. Currently, three different human diseases are associated with PIEZO1 missense mutations: (i) dehydrated hereditary stomatocytosis (OMIM 194380), an autosomal dominant hemolytic anemia caused by gain-of-function mutations[33,34],

(ii) lymphatic malformation-6 (OMIM 616843), an autosomal recessive condition caused by loss-of-function mutations[35,36], and (iii) serologically defined Er alloantibodies causing severe hemolytic disease of the fetus and newborn (OMIM 620207), caused by homozygous, compound heterozygous, or heterozygous mutations in the PIEZO1 gene[37]. In our case, phenotypically, the PBS male born from asymptomatic carrier parents has isolated PBS of moderate severity and thus far, at adult age he does not manifest any unique phenotypic features beyond PBS. He has no history of anemia, hydrops fetalis, or serologically defined Er alloantibodies although there is history of a sibling fetal loss (spontaneous natural abortion) which conceivably could be related. In review of the literature, a study published by Fotiou et al., 2015 identified one female patient with generalized lymphatic dysplasia and additional clinical features including 'prune' belly. Like our patient, this proband harbors compound heterozygous PIEZO1 mutations, two inherited from the mother and one inherited from the

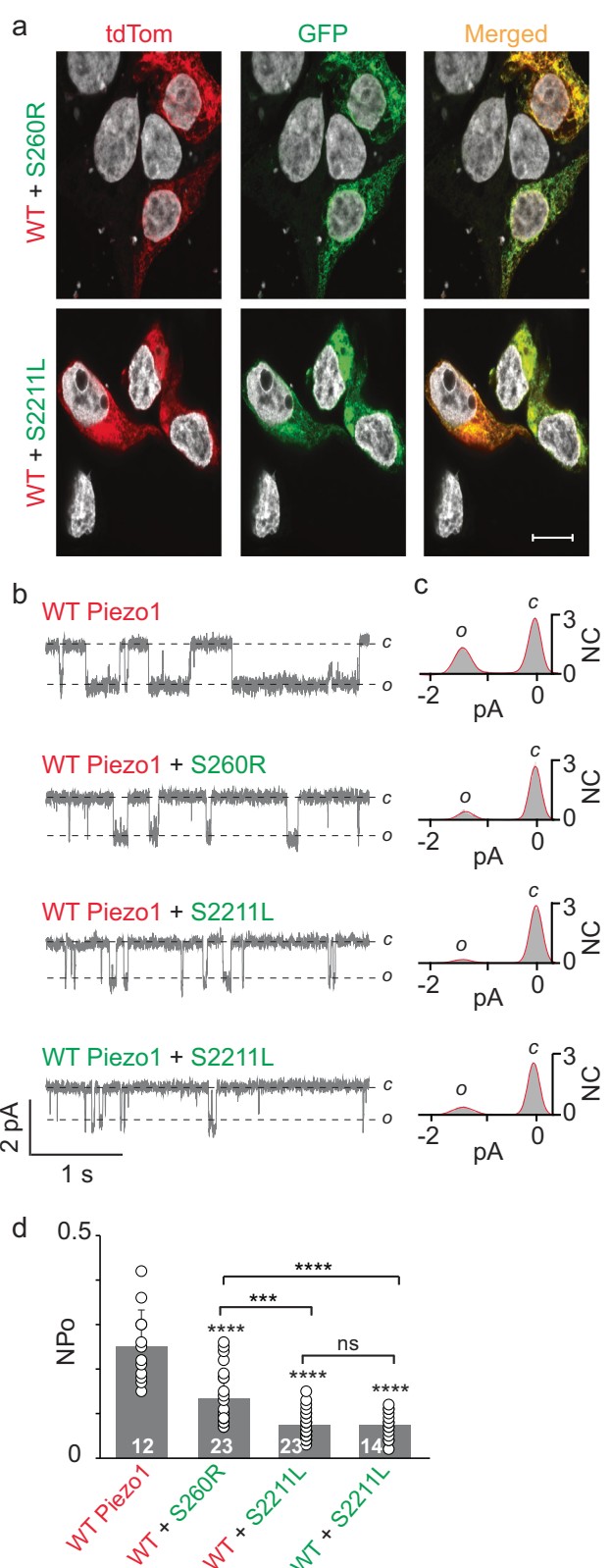

**Fig. 3 | Evaluation for loss-of-function effect of Piezo1 PBS mutations in HEK293T$^{\Delta P1}$ cells. a** Representative confocal images ($n = 12$) of HEK293T$^{\Delta P1}$ cells co-expressing WT-tdTom with either S260R-GFP (upper) or S2211L-GFP (lower) fused proteins. **b** Single-channel current recordings of WT and PBS variant proteins expressing WT-tdTom alone, WT-tdTom with either S260R-GFP or S2211L-GFP. WT-GFP with S2211L-GFP was used as control. **c** 30 s all-point current histograms of the single-channel recordings shown in (**b**). **d** Steady state NPo of WT-dtTom and PBS-GFP variants. Statistical analysis was performed by using One-way ANOVA where ***$p < 0.001$, ****$p < 0.0001$ and ns = not significant. tdTom = tdTomato, C = close, O = open, pA = picoamperes, NC = normalized counts, NPo = normalized open probability, calibration bar = 10 μm. Data points in (**d**) are represented as Mean ± SD.

'prune' belly may have phenotypic overlap with PBS. However, there are clear phenotypic differences, since PBS is typically in male patients with undescended testes and there is typically urinary tract dilation (megabladder, megaureter, hydronephrosis, etc). Fotiou et al., do not report any urinary tract phenotypes in this compound heterozygous female case. The PBS proband in our report has Supraventricular tachycardia (SVT) and his brother with unknown PIEZO1 variant status (no DNA available for study) has thoracic aortic aneurysm and dissection (TAAD). Although PIEZO1 has not been directly connected with SVT and TAAD, in mice this protein serves as a key cardiovascular mechanotransducer, maintaining normal cardiovascular function[39–43], suggesting a potential connection of the role of PIEZO1 mutations on the multisystem PBS phenotype.

At this point, most cases of PBS remain genetically undefined. In the last ten years, seven autosomal genes (MYOCD, CHRM3, HNF1β, ACTA2, ACTG2, STIM1 and MYH11) and one X-linked gene (FLNA) have been reported with potentially causal rare DNA variants, including structural variants, copy number variants, and single nucleotide variants[44–54]. Interestingly, FLNA, MYH11, ACTA2, and ACTG2 genes are involved in mechanosensation of the cell[55–58]. This reinforces the relevance of mechanosensors in bladder physiology and PBS. Overall, these studies suggest that PBS is not monogenic. PBS may be caused by mutations in multiple genes and the other causal genes remain to be identified and functionally validated[46]. Concerning our PBS proband reported here, his WES data has not identified exonic mutations in any of these 8 genes. However, in addition to the two mutations in PIEZO1, we identified 61 other genes with 61 rare, damaging variants identified by filtering criteria (Supplementary Table 1). PIEZO1 was the only gene manifesting compound heterozygosity and therefore, we focused our efforts on functionally characterizing these PIEZO1 variants.

Cryo-EM structure shows that mPiezo1 is a homotrimer, where three identical monomers assemble together to form a functional channel (Fig. 1b). Given our PBS proband is compound heterozygous, it is possible that in this case, the PIEZO1 trimer is a heterotrimer of G253R and S2195L and the stoichiometry of the mutant alleles may impact the trimeric protein function. Theoretically, four trimeric assemblies are possible after co-expression of PBS mutants: 1. S260R homotrimers, 2. S2211L homotrimers, 3. S260R:S2211L heterotrimeric assembly ratio of 2:1, or 4. S260R:S2211L heterotrimeric assembly ratio of 1:2. Functionally, we see evidence that the co-expressed mutants are different than their respective counterparts. Our data suggested that the co-expression of S260R and S2211L is significantly different than S260R expressed alone (Figs 2, 4 and 5). Functional heterotrimers have been reported when mPiezo1 and mPiezo2 were co-expressed or mutants of mPiezo1 were co-expressed[59,60]. Here, we employ similar methods to assemble heterotrimers between WT mPiezo1 and human PBS mutations. Although, we cannot assert structurally the occurrence of heteromeric channels of PBS mutants, it is possible that pressure-dependent NPo is further fine-tuned by the identity and number of mutant subunits in a PIEZO1 trimer.

Previous in vitro studies and mouse models have established the importance of mPiezo1 in urinary bladder physiology[18,19,61]. An aberrant

father. One of these variants L939M has been tested functionally to compare with WT Piezo1 activity[38]. The peak currents in response to stretch-activation were comparable between WT Piezo1 and L939M, however, modest rightward shift in the pressure-response curves were seen for the mutant[38]. Additionally, of these three variants, the L939M and F2458L are reported as "uncertain significance" and R2456C is not reported at all in ClinVar. It is possible that this single female case with

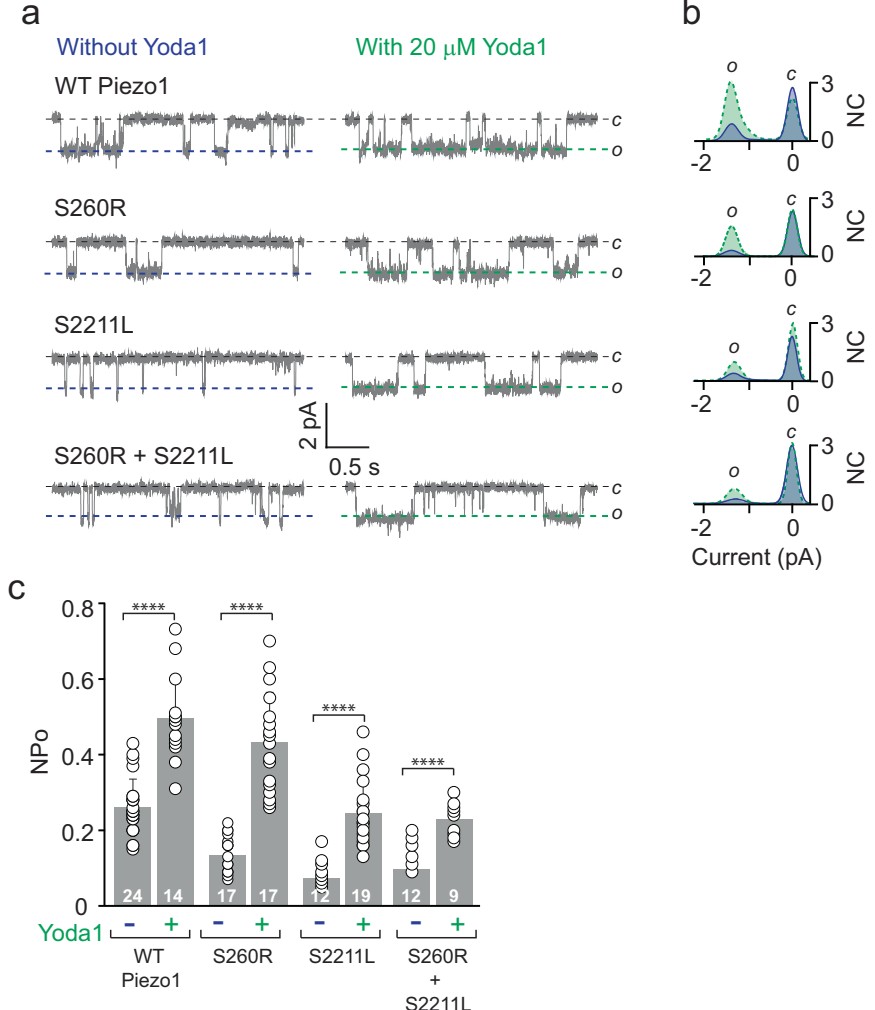

**Fig. 4 | Chemical modulation of PBS Piezo1 Variants by Yoda1. a** Representative single-channel current recordings of WT Piezo1 and PBS variant proteins S260R, S2211L and co-expressed S260R + S2211L either without Yoda1 (Left) or with 20 μM Yoda1 in the pipette (Right). The data was obtained from cell-attached patches at −60 mV and −30 mmHg applied pressure. *O* and *C* denote open and closed state of the channel. **b** Overlay of the corresponding all-point current amplitude histograms obtained from (**a**) in the absence (blue) and presence of Yoda1 (dash-green line). **c** Steady state NPo of WT Piezo1 and variant proteins with and without Yoda1 calculated from (**a**) data. Experimental replica numbers for each condition are shown in the bar graph (*n* > 9). Analysis was performed using unpaired two-tailed *t* test, where ****$p < 0.0001$. Data points in panel c are represented as Mean ± SD.

activation of mPiezo1 has been suggested to play a role in rodent bladder pathologies like partial bladder outlet obstruction and interstitial cystitis/bladder pain syndrome (IC/BPS)[61]. However, no human bladder disorder has been associated with dysregulation of PIEZO1. PIEZO1 and PIEZO2 are functionally distinct and are differentially expressed in tissues throughout the body[6,19]. Due to their distinct role during development, both mPiezo1 or mPiezo2 knockout (KO) mice die during embryogenesis, and at birth, respectively[22,62]. A Upk2-driven Cre conditional knockout of mPiezo1 and/or mPiezo2 was created by Dalghi et al.[18]. They found that despite mPiezo1 channels expression in all urothelial cells, urothelial mPiezo1-KO mice had limited bladder dysfunction phenotype. Piezo2 expression was limited to a small subset of superficial umbrella cells layers, and urothelial Piezo2-KO mice exhibited moderate bladder dysfunction. By crossing these mice, the dual urothelial Piezo1/2-KO mice had the most severe bladder phenotype[18]. A Hoxb8-driven Cre conditional knockout of Piezo2 from urothelial cells displayed similar phenotypes to the Upk2-driven Cre knockout animals, with higher bladder stretch thresholds, greater bladder pressure during micturition and attenuated urethral reflexes[5]. These studies suggested that Piezo1 function in the bladder urothelium has no significant impact and urothelial Piezo1 loss can be compensated by Piezo2[18].

The specific functional roles of PIEZO1 in mouse and human detrusor smooth muscle remain unclear, since a mouse Piezo1 conditional knockout in bladder SMCs has not been published. PIEZO1 is highly expressed in the bladder smooth muscle layer where Piezo2 is absent[19]. Thus, this information supports our hypothesis that PBS PIEZO1 loss-of-function mutations impact the proper activity of urinary bladder SMCs, and this functional loss cannot be rescued by PIEZO2. At the cell and organ level, we further theorize that the inability of bladder SMCs to properly perform mechanotransduction leads to chronic bladder distention, increased bladder compliance with poor contractility, and extracellular matrix deposition between detrusor SMCs. This PIEZO1 loss-of-function correlates with the PBS bladder phenotype.

In PBS, in addition to the altered urinary bladder function and form, the ventral abdominal wall is lacking and/or deficient in skeletal muscle cells, giving the affected individuals the characteristic wrinkled, flaccid "prune belly"[3]. This skeletal muscular deficiency is most profound in the regions overlying the distended bladder and/or the liver/kidneys where there is highest skeletal muscle mechanical strain[63,64]. Interestingly, Piezo1 is essential for maintaining skeletal muscle stem cell states, regulating protrusion length, and cell shape in

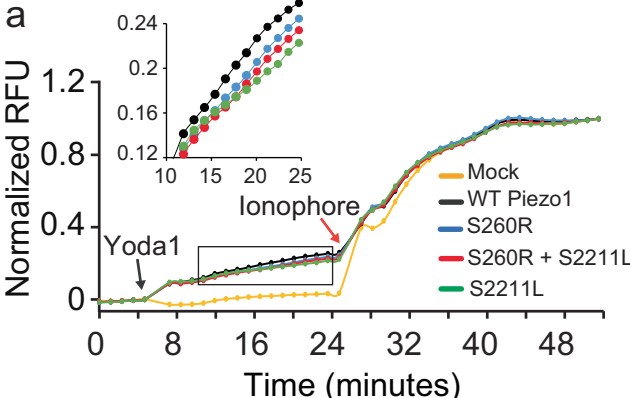

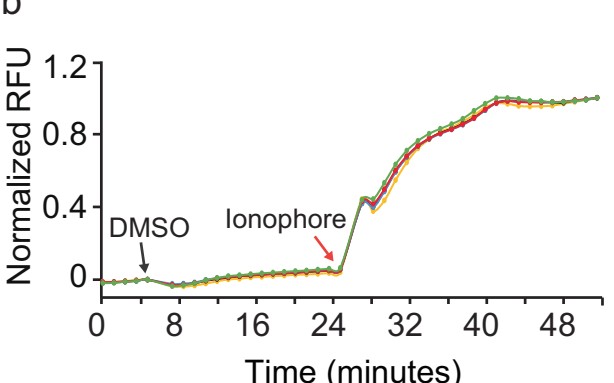

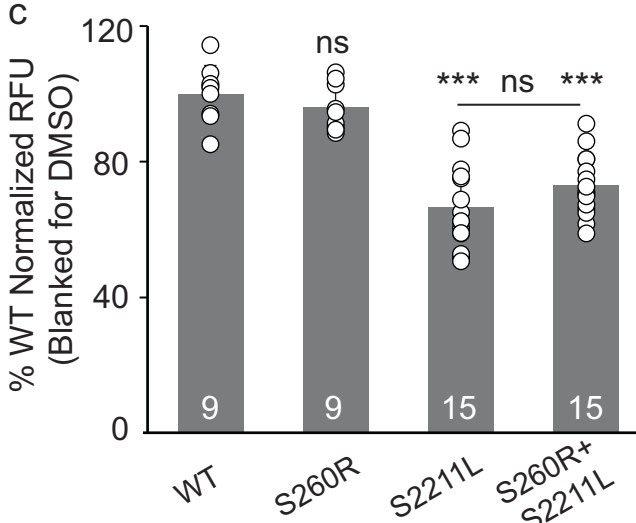

**Fig. 5 | Yoda1 specifically stimulates Piezo1-dependent Calcium Influx in HEK293T$^{\Delta P1}$ Cells expressing WT and PBS Piezo1 variants.** Fluorescent detection of intracellular calcium influx via Fluo-4 dye with or without Yoda1 of WT and PBS Piezo1 variants channel activity. Ionophore A23187 ($t$ = 24 min - red arrows) served as internal control to stimulate maximum non-specific calcium influx. **a** Addition of Yoda1 at 5 min (black arrow). Mock n = 8, WT $n$ = 9, S260R $n$ = 9, S2211L $n$ = 15, Co-expressed $n$ = 15. The insert shows the data from $t$ = 12 min to $t$ = 24 min. **b** Addition of DMSO at 5 min as a control (black arrow). Mock n = 4, WT $n$ = 8, S260R $n$ = 8, S2211L $n$ = 12, Co-expressed $n$ = 12. **c** Quantification of calcium influx indicated by normalized cell fluorescence induced by Yoda1. Data points reflect maximal activated signal prior to addition of ionophore A23187 ($t$ = 24 min). Mock = empty vector transfection and RFU = Relative fluorescence units. Statistical analysis was performed by using One-way ANOVA where ***$p$ < 0.001 and ns = not significant. Data points in (**c**) are represented as Mean ± SD.

both homeostatic and dystrophic muscles[64–66]. Additionally, it has been shown that Piezo1 expression increases during in vitro myogenesis while Piezo1 knockout decreased the myoblast fusion index in C2C12 cells, suggesting a critical role in skeletal muscle development and regeneration[65]. The studies also demonstrated that Yoda1 increased the distribution of responsive cells, promoted an active skeletal muscle stem cell state, and promoted myoblast fusion[67]. Therefore, in both smooth and skeletal muscle, Piezo1 expression and function are critically important during embryonic development and during adult life homeostasis, further supporting the etiological connection between the bladder and abdominal wall phenotype manifested in PBS patients.

Our loss of function PBS mutations only affect the NPo of the channel but has no effect on the single channel current (Fig. 2). Therefore, we used Yoda1 for a functional rescue. Yoda1 serves as a gating modifier, where it decreases the inactivation time but has no impact on the single channel current. Our in-vitro experiments (Figs 4 and 5) are proof of concept that Yoda1-like molecules may be used to alter the faulty protein function in vivo and fine-tune the Piezo1-dependent mechanical properties of the cells to near-WT levels. Each of the PBS mutations respond differently to Yoda1 stimulation, with S260R showing the most effective response to Yoda1 treatment, which is 96% of the WT, while S2211L is 66% and the co-expression is 72% of the WT Piezo1. Since its discovery in 2015[32], Yoda1 has served as an excellent drug for both in vitro and in vivo experiments to understand the physiology and function of Piezo1 channels[32]. However, Yoda1 has not been advanced to human clinical use for several reasons, including that Yoda1 has weak agonism, poor aqueous solubility, and lack of tissue-specific targeted drug delivery given PIEZO1's widespread cellular and tissue expression. Before clinical use becomes a reality for PIEZO1-related diseases, Yoda1-like molecules will need to be identified and/or chemically modified to minimize toxicity and to improve reliability, efficacy and potency as compared to existing Yoda1.

Based on our biophysical functional data we propose a working hypothesis: WT Piezo1 channels in the urinary bladder SMCs experience pressure in response to urine accumulation. The Piezo1 undergoes conformational change (from closed to open) upon pressure stimuli and allows ionic influx (Calcium and Sodium) down their electrochemical gradient. While PBS compound heterozygous mutant channels exhibit impaired pressure sensitivity and lower probability of opening in response to pressure. We demonstrated that the lower open probability of mutant channels results in lower influx of calcium and sodium which in return can affect many downstream signaling events including calcium dependent contraction of the bladder via MLC kinase pathway. This could be one of the reasons for underactive bladder in PBS patients. Our data also demonstrate that small molecule Yoda1 can increase the open probability of the mutant channels and overall ionic influx rescuing the phenotype of mutant channels.

Here, we identified a potential genetic basis for the congenital human disorder PBS by identifying an individual with compound heterozygous PIEZO1 point mutations. Upon detailed biophysical analyses in HEK293T$^{\Delta P1}$ cells, both mutations manifest gating defects in a loss-of-function manner. Notably, we demonstrated that the gating function of PIEZO1 mutations could be recovered by small molecule Yoda1, opening the door for future genetic studies and personalized medicine approaches to management of this rare disease.

## Methods
### Ethical statement
We prospectively enrolled individuals with PBS and their family members in our IRB-approved at UTSW (STU032011-174) and Nationwide Children's Hospital (STUDY00003343) Pediatric Genitourinary DNA Repository starting in 2001. All procedures were in accordance with the ethical standards of the relevant committees on human

experimentation. Available medical records from the study subject were retrospectively reviewed and in person and/or telephone interviews were performed to obtain medical, surgical, and family history and medical photographs. Given PBS is a multi-organ system congenital disorder, we calculated the clinical phenotype severity score using RUBACE (R: renal, U: ureter, B: bladder/outlet, A: abdominal wall, C: cryptorchidism, E: extra-genitourinary), a scoring system developed by our group to quantify PBS disease severity and categorize patients into isolated PBS, syndromic PBS or PBS-Plus groups[2].

## Whole exome sequencing

Genomic DNA from participants was extracted according to manufacturer's protocols from blood lymphocytes using the Puregene DNA isolation kit (Gentra/Qiagen) or from saliva (Oragene). Paired-end WES was performed at the UT Southwestern McDermott Next Generation Sequencing Core using the Illumina HiSeq2500. Library preparation was done using the Illumina SureSelect DNA Sample prep kit and captured with the Illumina SureSelect Exome Enrichment kit. Data processing and analysis were performed by the UTSW McDermott Center Bioinformatics group. Adaptor removal and sample demultiplexing were done using CASAVA and BWA was used for alignment to the human genome (GRCh38/hg19). Mapped reads were processed, sorted, and underwent duplicate removal using Samtools and PICARD, and GATK was used for quality control, including realignment around insertions and deletions and base quality score recalibration. Variant calling was performed using data training sets from the 1000 Genomes Project, Omni 2.5 M SNP microarray and HapMap phase 3.3. Variants were filtered for rare minor allele frequency (<0.001) and their potential functions were predicted using in silico-based computational analysis. Variants meeting at least five of the following criteria were selected as candidate functional variants: SIFT < 0.1, Polyphen2 > 0.9, CADD score >20, GERP > 4, MutationTaster = "DC", Mutation Assessor = "M" or "NE", and Vest >3.00. Rare and novel candidate functional variants were screened in ClinVar database for prior association with smooth muscle myopathy (SMM). Only variants that met these criteria were included for further analysis.

## Sanger sequencing

Variants identified from WES were confirmed using available proband and familial DNAs. DNAs were PCR-amplified utilizing Roche Kapa2G HotStart ReadyMix enzyme and variant-specific Sigma-Aldrich primers (Piezo-1_S2195L: Forward 5′- GGGCATCTGGTAGCAGTAGAG and Reverse 5′- TTTCCCATCAGCACTCGGG; Piezo-1_G253L: Forward 5′- ACATGCTCACCTCATAGCCG and Reverse 5′- ACCATTACTACCGC TGTGCC). After amplification in T100 Thermal Cycler (Bio-Rad), PCR products were washed and extracted by QIAquick Gel Extraction Kit. Final PCR DNAs were Sanger sequenced at UT Southwestern McDermott Center Sequencing Core.

## Evolutionary analysis

PIEZO1 protein sequences were derived from full-length NCBI accessions and genomic sequences. Human (*Homo sapiens*): NP_001136336.2, Chimpanzee (*Pan troglodytes*): XP_016785866.2, Rhesus (*Macaca mulatta*): XP_028696932.1, Rat (*Rattus norvegicus*): NP_001070668.2, Mouse (*Mus musculus*): NP_001032375.1, Chicken (*Gallus gallus*): XP_040537377.1, Cow (*Bos Taurus*): XP_015331270.2, Dog (*Canis lupus dingo*): XP_035572024.1, Frog (*Xenopus laevis*): XP 002933721.3, Zebra fish (*Danio rerio*): XP 696355.4. The protein sequences from different vertebrate species of PIEZO1 were aligned using ClustalW2 multiple sequence-alignment software.

## Piezo1 clone generation

PBS human mutations were mapped onto the mouse genome to derive the mouse mutations for transfection. The human Gly253Arg was mapped as murine Ser260Arg, and the human Ser2195Leu was mapped as murine Ser2211Leu. The full-length mouse Piezo1-GFP fused gene was synthesized using GenScript's Clone EZ service. Subsequent PBS point mutations clones for Piezo1 bearing S260R or S2211L mutants were generated from this parent clone, also using GenScript's Clone EZ. For the calcium flux assay (Fig. 5) and cell imaging assays, tag at amino acid 2422 was introduced and GFP was deleted to avoid fluorescence signal from fused GFP tagged. All nucleotide sequences were sequenced verified after every maxi prep DNA isolation and sequence verified by UT Southwestern Sequencing Core. Primers for sequencing were purchased from IDT and are as follows:

S260R (forward):5′-CTGGTGGTCCTGTCACTTTC-3′
S260R (reverse):5′-GCTCTGGCTGGTTAGTACAT-3′
S2211L (forward):5′-AGCCGAGAGACAGAGAAGAA-3′
S2211L (reverse):5′-CTCAGGACTGTACTGGCTAATG-3′
C-term w/2422-Myc tag (forward): 5′-TTCCCCATCTCTTCCCC AAG-3′
C-term w/2422-Myc tag (reverse): 5′- GGAAGATGAGCTTGGC GTATAG −3′

## Cell culture and transient transfection

HEK293T$^{\Delta P1}$ cells (PIEZO1 Knockout cell line) were purchased from ATCC (#CRL-3519). Cell line was authenticated as Piezo1 knock out by electrophysiology and calcium flux assays to verify no ion channel activity in each functional study, in the presence of small molecule Yoda1 (a chemical activator of Piezo1). Cell lines were confirmed to be mycoplasma-negative when cultures were started in the lab. HEK293T$^{\Delta P1}$ cells were maintained in DMEM (HyClone, SH3008101) supplemented with 10% Fetal Bovine Serum (FBS) (Millipore Sigma, F2422), penicillin streptomycin (Gibco, 15140122). HEK293T$^{\Delta P1}$ cells were transfected with 1 μg of Piezo1-myc, S260R-myc, or S2211L-myc, Piezo1-GFP, S260R-GFP, S2211L-GFP, or Piezo1-tdtom constructs. Transient transfection was performed using Lipofectamine 3000 (Invitrogen, L3000008) according to the manufacturer's instructions. All the recordings were performed between 16–20 h after transfection.

## Electrophysiology

Transfected HEK293T$^{\Delta P1}$ cells were visualized using Nikon eclipse Ti2 microscope and C11440 Orca-Flash 4.0 LT digital camera (Hamamatsu). Cell-attached recordings of mechano-activated currents in HEK293T$^{\Delta P1}$ cells expressing the WT Piezo1 and PBS point mutations were performed using experimental setup based on an Axopatch 200B amplifier and Digidata 1550B digitizer (Molecular Devices). Currents were acquired with pClamp 10.7 software and were recorded at a sampling frequency of 10 kHz. Recording patch pipettes of borosilicate glass (Sutter Instruments BF150-86-10) were pulled and fire-polished to a tip resistance of 4 to 6 MΩ. The bath solution contained (in mM): 140 KCl (Fisher Bioreagents, BP366-500), 10 HEPES (Gibco, 15630080), 1 MgCl2 (Millipore Sigma, 63069), 10 glucose (Sigma, G8270-1KG), pH 7.3 (pH adjusted with KOH {LabChem LC193702}). The pipette solution contained (in mM): 130 NaCl (Fisher Brand, S271-3), 5 KCl, 10 HEPES, 10 TEA-Cl (Acros Organics, 150901000), 1 CaCl2 (Millipore Sigma, 21115), 1 MgCl2 pH 7.3 (pH adjusted with NaOH). To observe Yoda1 effect, 20 μM Yoda1 (Tocris, 5586/10) was supplemented in the pipette solution. The bath electrode was grounded while the pipette electrode was active such that at positive applied potential, cations moved from pipette to bath (inward currents). Mechanical stimulation was applied via recording pipette using a high-speed pressure clamp system (ALA Scientific Instruments). Single-channel events are shown as downward deflection representing inward currents acquired at −60 mV (to match conventional direction of flow). The single-channel data was analyzed using Clampfit 10.7. The Normalized Open Probability was calculated by:

$$NPo = A_o / A_c + A_o$$

Where $A_o$ and $A_c$ are the areas under the curve of open and closed components of all-point histograms respectively. The areas under the curve of all-point histograms were calculated using Clampfit analysis program and via Gaussian fits to the data.

## Calcium flux assay

Fluo-4 AM NW dye and probenecid (Molecular Probes, 36206) were prepared according to manufacturer's instructions. The dye was further supplemented to a final 4 mM $CaCl_2$ concentration. Yoda1 was prepared as 10 mM stock in 100% DMSO (Molecular Probes, D12345). 24 h after HEK293T$^{\Delta P1}$ cells were seeded in 96-well plates (Corning, 3603) and transfected with Piezo1-myc, S260R-myc, or S2211L-myc constructs. Media was aspirated to reduce background and adherent cells were loaded with dye/probenecid solution for 40 min at 37 °C and 8% $CO_2$. On loading, baseline fluorescence was measured for 5 min, and activation by 20 μM Yoda-1 (in 0.2% DMSO) or 0.2% DMSO alone was measured for 18 min. Maximal calcium signal was determined by addition of A23187 ionophore (Millipore Sigma, C7522) and assayed for 25 min. All Relative Fluorescence Unit (RFU) measurements were performed on a BioTek Synergy H1 at 37 °C with excitation 488 nm and emission 518 nm, and 70 s between reads. Normalized RFU = (signal RFU - baseline RFU) / (maximal post-A23187 - baseline RFU). Signal RFU = mean well RFU at $t = 24$ min, baseline RFU = mean well RFU at $t = 5$ min, maximal post-A23187 = mean signal at $t = 52$ min. Data point represents $n = 9$ to 15 normalized wells. Each well is plated, transfected, treated, and assayed independently, and as such, represents individual experiments.

## Immunofluorescence

For immunofluorescence, HEK293T$^{\Delta P1}$ cells were either grown on poly-D-lysine (Gibco, A3890401) coated glass coverslips or poly-D-lysine coated glass-bottom wells (ibidi, 80807). To visualize Piezo1-tdTomato and/or Piezo1-GFP, at 24 h after transfection, cells were fixed with 4% paraformaldehyde for 15 min at room temperature and washed 3 times with 1X Phosphate buffered saline (PhosBS). Nuclear staining for all constructs was performed by incubating cells in a 1:5000 DAPI to 1X PhosBS solution for 1 min. The cells were then washed three times for 1 min each with 1X PhosBS. Cells were mounted using a drop of Prolong Diamond Antifade Mountant (Thermo Fischer) placed on a Histobond Adhesive Slide (StatLab), and the slide is allowed to cure overnight at room temperature. The cells were imaged with a Zeiss LSM880 confocal laser scanning microscope at 63x magnification using a Zeiss Plan Apo 63x/1.40 oil objective.

To qualitatively assess surface expression, Piezo1-myc transfected HEK293T$^{\Delta P1}$ cells were grown in poly-D-lysine (gibco, A3890401) coated glass-bottom wells (ibidi, 80807) and were fixed with 4% paraformaldehyde (Thermo scientific, J61899.AK) for 15 min at room temperature and washed 3 times with 1X PBS (gibco, 10010023). Cells were next blocked with 10% normal horse serum (Sigma-Aldrich, H0146-5ML) and 10% BSA (Fisher BioReagents, BP9706100) and incubated with Alexa Fluor 555 conjugate Myc-antibody overnight at 4 °C (Cell Signaling Technology, 3756 S). The Myc-antibody was validated by using system null for Myc-fusion of Piezo1. The cells were washed with 1X PBS to remove unbound, excess antibody. Nucleus staining for all constructs were performed by incubating cells in a 1:5000 DAPI (Sigma-Aldrich, D9542-10MG) to 1X PBS solution for 1 min. The cells were washed 3 times with 1X PBS. The cells were then imaged, capturing the surface expression of Piezo1-myc, with a Zeiss LSM880 confocal laser scanning microscope at 63x magnification using a Zeiss Plan Apo 63x/1.40 oil objective. To obtain permeabilized images, the same samples were then blocked with 10% normal horse serum, 10% BSA, and 0.4% Triton-x100 and again incubated with Alexa Fluor 555 conjugate Myc-antibody overnight at 4 °C, and washed with 1X PBS. The cells were imaged again, capturing the internal expression of Piezo1-myc.

## Statistics

All-point current histograms of the single-channel data (ranging from 15 to 30 s stretch) were constructed in Clampfit. The single-channel current values were extracted after fitting the Gaussian function to the data. Each mean value is an average of at least 12 or at most 24 individual patch recordings. 8–12 individual coverslips with transfected cells, either WT Piezo1 or mutants, were used to acquire single channel data. NPo of the channel was calculated exclusively from the records, where at least 10 s of data was recorded both before and after pressure application. The number of channels in the patch is determined by fitting a Gaussian curve to all-point current histograms to individual open peak. At least a 15–30 s stretch of recordings were analyzed to observe for consecutive channel openings. Group data (conductance bar graphs) are presented as Mean ± Standard Deviation where $^*p < 0.05$; $^{**}p < 0.01$, $^{***}p < 0.001$, $^{****}p < 0.0001$, not significant (ns) $p > 0.05$. Unpaired two-tailed $t$ tests were used for comparison between the two groups in question. For multiple group comparisons, a one-way ANOVA with Bonferroni correction was performed.

## Study approval

Written informed consent for DNA, medical records and photography was obtained from the proband prior to performing the studies. This study was approved by the UT Southwestern IRB (IRB #STU032011-174). In vitro functional studies were conducted under the approval of the Institutional Biosafety Committee of UT Southwestern (IBC # RDSR-2021-074).

## Reporting summary

Further information on research design is available in the Nature Portfolio Reporting Summary linked to this article.

## Data availability

The reagents, Piezo1 clones, protocols and data are available upon request, and are also included in the paper. The reference human genome used to align and map the whole exome sequence was the GRCh38/hg19. The whole exome sequencing data of this research are available under restricted access at the database of Genotypes and Phenotypes (bdGaP), Accession ID: phs003475.v1.p1. The reason for the restricted access to the data is to protect individual genetic information. Qualified investigators must submit an online Data Access Request [https://dbgap.ncbi.nlm.nih.gov/aa/wga.cgi?page=login] with more information found here [https://www.ncbi.nlm.nih.gov/projects/gap/cgi-bin/GetPdf.cgi?document_name=GeneralAAInstructions.pdf]. Once access is granted, the data will be available to researchers for the duration agreed upon within the data access agreement, with the possibility of extension upon request. Source data are available as a Source Data file. Source data are provided with this paper.

## Code availability

The script for the whole exome sequencing analysis is available in the GitHub at https://github.com/medforomics/process_scripts/tree/master.

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

## Acknowledgements

The authors warmly thank the wonderful PBS family who contributed and express deepest gratitude to the Prune Belly Syndrome Network (www.prunebelly.org) for their invaluable help. L.A.B. is supported by a grant from the National Institutes of Health (RO1 DK127589 and RO1 DK105068), the Seay Endowment for Pediatric Urologic Research, and startup funds from Nationwide Children's Hospital. N.G.A. is supported by an American Urology Association Urology Care Foundation Research Scholars Award. R.S is a W.W. Caruth, Jr. Scholar in Biomedical Research and supported by a grant from the National Institute of Health (RO1 GM142024) and UT Southwestern Medical Center Endowed Scholar Program. We are grateful to the UT Southwestern McDermott Sequencing Center (WES analysis) and acknowledge the assistance of the UT Southwestern Tissue Resource, a shared resource at the Simmons Comprehensive Cancer Center supported in part by the National Cancer Institute under award number 5P30CA142543.

## Author contributions

N.G.A. designed the research studies, conducted sequencing experiments, acquired clinical data, acquired, and analyzed data, and drafted, wrote, and revised the manuscript. E.D.N. designed the research studies, conducted electrophysiology experiments, acquired, and analyzed data. D.T. conducted biochemistry experiments, acquired, and analyzed data. T.J.E. conducted sequencing experiments and acquired data. O.W.O. conducted microscopy experiments and acquired data. M.Y. acquired clinical data; A.N.F. conducted sequencing experiments and acquired data; N.P. acquired clinical data; J.M. analyzed bioinformatics data; B.C. analyzed bioinformatics data; L.A.B. initiated the project, identified, recruited, and collected samples from the human subjects, design experiments, gathered the research team, designed the research studies, supervised the experiments, interpreted the data, and drafted and wrote the manuscript. R.S. designed the research studies, supervised the experiments, interpreted data, and drafted and wrote the manuscript. The method used in assigning the co–first authorship and co-senior authorship order was based on the relative contributions of the authors to the manuscript.

## Competing interests

The authors declare no competing interests.
