## [Peer Review File · Nature Communications]

PIEZO1 loss-of-function compound heterozygous mutations in the rare congenital human disorder Prune Belly Syndrome.Reviewers' Comments:

Reviewer #1:

Remarks to the Author:

This interesting study from Amado et al., links PIEZO1 variants to the rare multi-system congenital disorder prune belly syndrome (PBS). The authors identify compound heterozygous variants in *PIEZO1* in a proband with PBS and using single channel patch-clamp recordings show that these variants influence the gating of PIEZO1, which would add to the spectrum of disorders known to be caused by PIEZO1 dysfunction. However, there are numerous very important points that the authors must address prior to publication:

1. 1 proband with many possibly/probably damaging variants

The first striking thing is the unequivocal way in which the authors ascribe causation to PIEZO1 with only a single proband and at that, a proband with ~60 other variants of interest (e.g. "we identified a new genetic basis for the congenital human disorder PBS" and "the cellular pathophysiological phenotype can be rescued by the small molecule, Yoda1"). Regardless of the compound heterozygosity of *PIEZO1*, without a specific cellular mechanism or animal model, ascribing causation to these *PIEZO1* variants is difficult even with the beautifully presented single channel data. It is necessary to be much more circumspect with the language.

2. Dominant negative

I find it strange the mutations are discussed to have a dominant negative effect. If we look at dominant negative variants in other ion channels there are few examples of dominant negative mutations requiring compound heterozygosity usually these are severe and will instigate disease alone – as per the dominant negative status. How do you explain the lack of penetrance in the parents if the variants are dominant negative? Are there other PIEZO1 variants reported to be dominant negative? Intimately related to this point are details regarding the transfection protocol. To understand figure 3 that purports a dominant negative effect, it is important to include the exact protocols for transfection when transfecting multiple mutants into the same cell – this includes the exact times and quantity of DNA used. Presumably we are not looking at X mcg of WT and comparing to X mcg WT + X mcg mutant DNA? Without this information it is difficult to see how the data presented in Figure 3D supports that S260R is a dominant negative variant.

3. Macro-currents

The lower NPo of the mutants at -30 mmHg shown in Figure 2 is clearly consistent with channels less sensitive to mechanical force. However, the only supporting evidence I can find for this conclusion is: Line 163 "Specifically, NPo increased after pressure application in WT Piezo1, S260R expressed alone, and S2211L expressed alone by 4, 1.5, and 3-fold, respectively." However, this data is not shown and has no statistics associated with it. It is absolutely necessary to show macro-currents from cell-attached patches and calculate the pressure sensitivity and document the p50 of the variants compared to the WT. This will also enable an assessment of inactivation kinetics as enhanced inactivation could also conceivably result in an LOF phenotype where NPo at a set pressure is lower than WT. In addition, from Fig 3 it seems clear shorter open dwell times is what drives the decreased NPo in the mutants which could also be consistent with an influence on inactivation. Either detailed kinetic analysis of the single channels (e.g open/closed dwell time, state models) is required or analysis of macro-currents. Without this, the biophysical characterization of the mutants is incomplete, and the mechanism of reduced activity remains unclear.

4. Statistics

Data presented in Figure 2E and Figure 3D should be analysed using one-way ANOVA not multiple t-tests.

Minor

- Piezo1, *PIEZO1*, PIEZO1. These should be uniform throughout with the correct and

appropriate use of italics when referring to the gene.

- Line 58 "cellular physical forces (pressure, stretch, and tension)" in this context pressure/stretch means nothing, shear and tensile or compressive stress are what the channel responds to.

- Line 60-64 is poorly referenced and the words in parenthesis are not required (permeation/conductance).

- Figure 4D presents identical data to 4C, it is therefore superfluous – amend.

- The differences in Figure 5A seem small in comparison to the averaged data in Figure 5C please include an expanded inset of the data in Figure 5A to allow the reader to interrogate the raw data. Also, what do the n-numbers represent and does Fig5A represent averaged data or one example representative experiment? These details seem to be missing. What DNA constructs were used for these experiments?

- Figure 6 provides little to no visual benefit to the paper and should be removed.

- Line 288-296 seems to be a section that should be in the introduction it adds little to the discussion which is already verbose.

- Typographical or grammatical errors:

Line 240 "9No"

Line 468 "To observe Yoda1 effect, 20 mM Yoda1 was supplemented in the pipette solution."

Line 762 "-30mm Hg" this error is in multiple places.

Line 804 "bar=10um"

Line 806 Figure 4 "20uM Yoda1"

Reviewer #2:

Remarks to the Author:

This is an interesting study reporting, for the first time, a functional connection between Piezo1 dysfunction and prune belly syndrome. PBS is a rare disorder with no known genetic cause. The authors show compelling in vitro cell-attached electrophysiology data suggesting that a partial loss of function could contribute to PBS. This is a major advance in understanding the genetics of PBS for which there is currently no cure. The methodology is sound, and the conclusions are supported by data. The observation that Piezo1 activation restores channel function opens up a promising therapeutic avenue to PBS treatment.

Minor comments

60 and other places in the manuscript: the authors use Piezo1 or PIEZO1 while referring to the same protein? Please correct for consistent nomenclature.

63 current is not the same as conductance, please correct.

65 RBC = red blood cells?

Figure 2: please add exemplar traces with the pressure applied.

Figure 2: is there a difference between the black bars in 2C and panel 2E?

Figure 2, 3, 4, 5, S2, S4: indicate in the legend statistical tests used.

Figure 2E legend: can you elaborate how NPo was derived from AUC?

Figure 5: are the RFUs normalized (e.g. divided by the baseline value) or baseline-subtracted?

Figure 5A, B: please clarify what these traces are: means from n cells with errors?

Figure 5C: please clarify where in 5A these data were measured at? The differences between the SL and SL/SR compared to WT seems much larger than what can be seen in 5A.

173 "this clearly shows that PBS point mutations have intact single channel conductance" – this needs to be corrected as conductance has only been measured at -60mV (while slope conductance could be different), so this has to be indicated (conductance at -60 mV). This has to be indicated whenever the authors mention no change in conductance.

We thank the reviewers for taking their time to read and provide critical constructive comments. We have addressed below (in red) point by point answers to reviewers' questions and suggestions, and where necessary mentioned the line numbers for the changes made in the main text file.

Reviewer #1 (Remarks to the Author):

This interesting study from Amado et al., links PIEZO1 variants to the rare multi-system congenital disorder prune belly syndrome (PBS). The authors identify compound heterozygous variants in *PIEZO1* in a proband with PBS and using single channel patch-clamp recordings show that these variants influence the gating of PIEZO1, which would add to the spectrum of disorders known to be caused by PIEZO1 dysfunction. However, there are numerous very important points that the authors must address prior to publication:

We thank the reviewer for reading the paper in detail and for their generous comments. Please see the point-by-point response below.

1. 1 proband with many possibly/probably damaging variants
The first striking thing is the unequivocal way in which the authors ascribe causation to PIEZO1 with only a single proband and at that, a proband with ~60 other variants of interest (e.g. “we identified a new genetic basis for the congenital human disorder PBS” and “the cellular pathophysiological phenotype can be rescued by the small molecule, Yoda1”). Regardless of the compound heterozygosity of *PIEZO1*, without a specific cellular mechanism or animal model, ascribing causation to these *PIEZO1* variants is difficult even with the beautifully presented single channel data. It is necessary to be much more circumspect with the language.

We agree with the reviewer that the language should be softened. Concerning the ~60 other variants we identified in our PBS case, none of them were compound heterozygous, none were in genes implicated in muscle development or function and none have undergone validation or functional testing. For these reasons, we have not called these additional 61 variants “variants of interest”.

We agree that the presence of only one proband weakens the evidence for causation. In review of the literature, a study published by Fotiou et al, 2015 identified one female patient with Generalized lymphatic dysplasia and additional clinical features including ‘prune’ belly. This proband harbors compound heterozygous PIEZO1 mutations, two inherited from the mother and one inherited from the father. These variants have not been tested *in vitro* or *in vivo* for functional significance. Of these three variants, one is reported as “uncertain significance” and one is not reported at all in ClinVar. It is possible

that this single female case with 'prune' belly may have phenotypic overlap with prune belly syndrome. However, there are clear phenotypic differences, since PBS is typically in male patients with undescended testes and there is typically urinary tract dilation (megabladder, megaureter, hydronephrosis, etc). Fotiou et al do not report any urinary tract phenotypes in this compound heterozygous female case. Although this may suggest an additional case of Prune belly syndrome with PIEZO1 mutation, the lack of clinical phenotypes and functional data does not give us sufficient confidence to initially add it to the paper, but now we have added this information to the Discussion (line 259 to 270).

As recommended, we have softened our wording in several locations in the paper (line 34, 35, 148, 397, and 401) making it more circumspect.

2. Dominant negative I find it strange the mutations are discussed to have a dominant negative effect. If we look at dominant negative variants in other ion channels there are few examples of dominant negative mutations requiring compound heterozygosity usually these are severe and will instigate disease alone – as per the dominant negative status. How do you explain the lack of penetrance in the parents if the variants are dominant negative? Are there other PIEZO1 variants reported to be dominant negative? Intimately related to this point are details regarding the transfection protocol. To understand figure 3 that purports a dominant negative effect, it is important to include the exact protocols for transfection when transfecting multiple mutants into the same cell – this includes the exact times and quantity of DNA used. Presumably we are not looking at X mcg of WT and comparing it to X mcg WT + X mcg mutant DNA? Without this information it is difficult to see how the data presented in Figure 3D supports that S260R is a dominant negative variant.

Thank you for your comments and you have made valid points. We agree with Reviewer #1 that our data do not prove dominant negative effect since we cannot control the amount of plasmid DNA taken by the cell. In the heterozygous parents, we believe the WT allele assures normal PIEZO1 function. We also clarified DNA quantities used and co-transfection times in the methods section (lines 486-488). Although we simultaneously co-transfected the same mcg amount of DNA and assayed 16-20h after transfection, in our experimental design, the cell DNA intake and expression is not controlled and could be different cell to cell.

While there are gain-of-function and loss-of-function PIEZO1 mutations reported in the literature, to the best of our knowledge dominant negative PIEZO1 mutations have not been reported yet.

We have revised the paper by deleting the term “dominant negative” and have used more appropriate language, such as loss of function (lines 31, 195, 196, 400, 826, and 827).

3. Macro-currents

The lower NPo of the mutants at -30 mmHg shown in Figure 2 is clearly consistent with channels less sensitive to mechanical force. However, the only supporting evidence I can find for this conclusion is: Line 163 “Specifically, NPo increased after pressure application in WT Piezo1, S260R expressed alone, and S2211L expressed alone by 4, 1.5, and 3-fold, respectively.” However, this data is not shown and has no statistics associated with it. It is absolutely necessary to show macro-currents from cell-attached patches and calculate the pressure sensitivity and document the p50 of the variants compared to the WT. This will also enable an assessment of inactivation kinetics as enhanced inactivation could also conceivably result in an LOF phenotype where NPo at a set pressure is lower than WT. In addition, from Fig 3 it seems clear shorter open dwell times is what drives the decreased NPo in the mutants which could also be consistent with an influence on inactivation. Either detailed kinetic analysis of the single channels (e.g open/closed dwell time, state models) is required or analysis of macro-currents. Without this, the biophysical characterization of the mutants is incomplete, and the mechanism of reduced activity remains unclear.

We apologize for not clarifying this point in the text. First, the 4-fold, 1.3-fold and 3-fold increase is derived from the data analyzed in Figure 2C. We have now added data in figure 2A before pressure application (grey traces) and after pressure application (red traces). When open probability is compared before and after the pressure application, within WT we see a 4-fold increase, and in mutants 1.3-fold and 3-fold respectively. No increase in open probability (before and after pressure) is observed when both the mutants were expressed together (in same DNA quantities).

We would like to show macro currents but consistent with our loss of function data, the mutants do not exhibit macro-currents in cell attached patches, as we can observe in WT Piezo1. At best, we have 3-4 consecutive channel openings in a steady state fashion but not sufficient to perform meaningful analysis of inactivation kinetics and p50 at various pressures for the mutants. Hence, we heavily relied on single-channel analysis.

In line with the reviewers' suggestion, we performed mean open time analysis and added as a new Panel in figure 2 (Figure 2F). We see a clear decrease in mean open time for both the mutants either expressed alone or co-expressed, in comparison to WT, when analyzed via dwell time histograms (presented as a new Supplementary Figure 3) (Line 185 to 193). Unfortunately, close time analysis is still tricky because the open probability/mean open time is very low. Even when we observe only 1 channel open in a 30 second record and no two consecutive openings were observed, we cannot be sure that it is one channel open and closing or two channels open and closing at different time, which will skew our mean close time analysis. Therefore, we did not present the mean

close times. We think mean open time in conjunction with open probability is the suitable way to suggest loss of function traits for mutants.

4. Statistics

Data presented in Figure 2E and Figure 3D should be analyzed using one-way ANOVA not multiple t-tests.

We thank the reviewer for the suggestion and now have performed one-way ANOVA and added it in legends for figure 2, 3, and 5 (lines 824-826, 835-836, and 867-868). One-way ANOVA testing found similar significant differences than the multiple t- tests and did not change the conclusions.

Minor

- Piezo1, *PIEZO1*, PIEZO1. These should be uniform throughout with the correct and appropriate use of italics when referring to the gene.

We apologize for this confusion and for not making it clear in the main text. The nomenclature in the PIEZO field is Mouse Piezo1 protein is not capitalized (mPiezo1), human PIEZO1 protein is capitalized. The human gene is in capitalized italics *PIEZO1*. We have now corrected it throughout the manuscript and explained it in the main text.

- Line 58 “cellular physical forces (pressure, stretch, and tension)” in this context pressure/stretch means nothing, shear and tensile or compressive stress are what the channel responds to.

We agree with the reviewer and now have corrected it on the line number 57.

- Line 60-64 is poorly referenced and the words in parenthesis are not required (permeation/conductance).

We agree with the reviewer’s corrections. We have added references number 11 to 14 for Line 60-64 (line 63). We have deleted (permeation /conductance) (line 61-62).

- Figure 4D presents identical data to 4C, it is therefore superfluous – amend.

We agree with the reviewer and now have deleted panel 4D. However, we added a comment in the main text lines 229 to 233 regarding panel 4D comparison. Similarly, we performed One-way ANOVA on this old figure 4D data and added this information to Results section lines 229 to 233. “No significant differences were found when WT Piezo1

without Yoda1 treatment was compared with S2211L expressed alone or co-expressed with S260R in the presence of Yoda1.”

- The differences in Figure 5A seem small in comparison to the averaged data in Figure 5C please include an expanded inset of the data in Figure 5A to allow the reader to interrogate the raw data. Also, what do the n-numbers represent and does Fig5A represent averaged data or one example representative experiment? These details seem to be missing. What DNA constructs were used for these experiments?

We thank the reviewer for helpful suggestions. We have now included inset in Fig 5A to allow the reader to see the differences in the raw data. Please see revised Figure 5A.

We have revised the methods section to address your questions (line 529 to 534). Figure 5A plots represent averaged data collected from 9 to 15 normalized wells. Each well is plated, transfected, treated, and assayed independently, and as such, represents individual experiments. We use myc-tag mPiezo1 (WT or PBS mutants) and we have added the DNA constructs information in the methods (line 521)

- Figure 6 provides little to no visual benefit to the paper and should be removed.

We agree with the reviewer’s suggestion and now have removed figure 6 from the paper.

- Line 288-296 seems to be a section that should be in the introduction, it adds little to the discussion which is already verbose.

We agree and have edit and moved this point to the introduction (line 66 to 69).

- Typographical or grammatical errors:

Line 240 “9No”

Line 468 “To observe Yoda1 effect, 20 mM Yoda1 was supplemented in the pipette solution.

Line 762 “-30mm Hg” this error is in multiple places.

Line 804 “bar=10um”

Line 806 Figure 4 “20uM Yoda1”

These typographical errors are now corrected (lines 275, 503, 811, 838, and 844).

Reviewer #2 (Remarks to the Author):

This is an interesting study reporting, for the first time, a functional connection between Piezo1 dysfunction and prune belly syndrome. PBS is a rare disorder with no known genetic cause. The authors show compelling in vitro cell-attached electrophysiology data suggesting that a partial loss of function could contribute to PBS. This is a major advance in understanding the genetics of PBS for which there is currently no cure. The methodology is sound, and the conclusions are supported by data. The observation that Piezo1 activation restores channel function opens up a promising therapeutic avenue to PBS treatment.

We thank the reviewer for reading the paper in detail and for their generous comments. Please see the point-by-point response below (in red).

Minor comments

60 and other places in the manuscript: the authors use Piezo1 or PIEZO1 while referring to the same protein? Please correct for consistent nomenclature.

We apologize for this confusion and for not making it clear in the main text. The nomenclature in the PIEZO field is Mouse Piezo1 protein is not capitalized (mPiezo1), Mouse Piezo1 gene is not capitalized and italic (*mPiezo1*), human PIEZO1 protein is capitalized. The human gene is in capitalized italics *PIEZO1*. We have now corrected it throughout the manuscript and explained it in the main text.

63 current is not the same as conductance, please correct.

We have now corrected it on line 61-62.

65 RBC = red blood cells?

We have defined RBC= red blood cells in the main text (line 64)

Figure 2: please add exemplar traces with the pressure applied.

Although we record data before and after applied pressure, previously we only showed traces with the applied pressure. We have now added data in figure 2A before pressure application (grey traces) and after pressure application (red traces) (lines 813-814)

Figure 2: is there a difference between the black bars in 2C and panel 2E?

No, they are the same data presented differently. In 2C we are comparing the WT and mutants before and after pressure. In Figure 2E WT vs mutants are compared only with the pressure applied. We have clarified this in the Figure 2E legend (line 824).

Figure 2, 3, 4, 5, S2, S4: indicate in the legend statistical tests used.

The statistical test used is now indicated in the legends of each figure (lines 824-826, 835-836, 848-851 and 867-868).

Figure 2E legend: can you elaborate how NPO was derived from AUC?

Apologies for the confusion in the legends. The open probability is calculated via Clampfit analysis that uses the formula.

$$NPO = A_o / (A_o + A_c)$$

Where A_o and A_c are the areas under the curve of open and closed components of all-point histograms respectively. The areas under the curve of all-point histograms were calculated using Clampfit analysis program and via Gaussian fits to the data.

We have clarified it in methods section under Electrophysiology (lines 509 to 515).

Figure 5: are the RFUs normalized (e.g., divided by the baseline value) or baseline-subtracted?

RFU is baseline subtracted and divided by maximal post-A23187 (ionophore) signal to normalize to maximal signal. We have clarified this point in figure 5 legend (line 867) and in methods section (lines 529 to 534).

Figure 5A, B: please clarify what these traces are: means from n cells with errors?

Figure 5A and 5B plots represent averaged data collected from the indicated number of wells (n #) from two different plates which were then combined as one data set. Cells were plated to be at near confluency on the day of assaying. Each well is plated, transfected, treated, and assayed independently, and as such, represents individual experiments. We have clarified this point in methods and in figure legends (lines 861 to 864).

- 5A n#: Mock = 8, WT = 9, S260R = 9, S2211L = 15, Co-expressed = 15
- 5B n#: Mock = 4, WT = 8, S260R = 8, S2211L = 12, Co-expressed = 12

Figure 5C: please clarify where in 5A these data were measured at? The differences between the SL and SL/SR compared to WT seems much larger than what can be seen in 5A.

The data in 5C is from the last/maximal data point of Yoda1 treatment, prior to addition of ionophore A23187. The differences are because of the inclusion of maximal post-A23187 signal. We have now included an inset in Figure 5A to address both Reviewer 1 and Reviewer 2 comments. In addition, the data in 5C represents not just the normalized data (that seen in 5A and 5B) but is further processed as % total of normalized WT values from 5A (lines 529 to 534).

173 “this clearly shows that PBS point mutations have intact single channel conductance” – this needs to be corrected as conductance has only been measured at -60mV (while slope conductance could be different), so this has to be indicated (conductance at -60 mV). This has to be indicated whenever the authors mention no change in conductance.

Thank you for correcting this point. We have now used -60 mV wherever we mentioned conductance of a channel.

Reviewers' Comments:

Reviewer #1:

Remarks to the Author:

Overall, the edits and changes to the manuscript are good but there are still some things that require addressing.

1. "In review of the literature, a study published by Fotiou et al, 2015 identified one female patient with Generalized lymphatic dysplasia and additional clinical features including 'prune' belly. This is indeed very supportive of the current findings!
2. "While there are gain-of-function and loss-of-function PIEZO1 mutations reported in the literature, to the best of our knowledge dominant negative PIEZO1 mutations have not been reported yet." A quick search reveals several articles that discuss and indeed show the possibility of dominant negative PIEZO1 variants (PMID: 32251670 PMID: 34489534 PMID: 34867393).
3. "This proband harbors compound heterozygous PIEZO1 mutations, two inherited from the mother and one inherited from the father. These variants have not been tested in vitro or in vivo for functional significance."

At least one of these variants has been tested in vitro PMID: 34489534

4. "No increase in open probability (before and after pressure) is observed when both the mutants were expressed together (in same DNA quantities)."

I still find it hard to understand why higher pressures are not used to assess NPo over a range of pressures. This would strongly support the idea that these channels have reduced sensitivity to mechanical force and given the point below I think it becomes even more important.

5. "We would like to show macro currents but consistent with our loss of function data, the mutants do not exhibit macro-currents in cell attached patches, as we can observe in WT Piezo1."

This sentence for mutants that are trafficked equally as well as WT is worrying. This should be clearly noted in the text as a reason for not showing macro-currents and Figure S4 which shows similar labelling of Myc-tagged mutants to WT, which is now an even more key experiment, should be included in Figure 3 (preferably with some form of quantification). It's even more difficult to understand why so few channels are encountered given the influence of Yoda on these mutants. If they have a right shift in their pressure response curve and express at identical levels to WT it's hard to understand why macro-currents can't be recorded unless they have additional effects on inactivation. This should be discussed in the manuscript.

Minor:

Some of the original errors I spotted have not been corrected!! - Figure 4 still says uM & Line 845 mm Hg

Line 166 assays

Line 276 there are data linking PIEZO1 to aortic aneurysm (PMID: 35082286) and structural abnormalities of the aortic valve (PMID: 32251670).

Line 219 punctuation

Line 260 hydrops fetalis is superfluous.

Line 400 "and behave as compound heterozygous mutations," doesn't make sense. (Also line 215)

Line 861 n 9

Reviewer #2:

Remarks to the Author:

I am fully satisfied with the revised version. All queries have been addressed.

We Thank the editor and the reviewers for taking time to read the first revised manuscript. We are very thankful for providing us with the opportunity to revise the manuscript with textual changes as suggested by reviewer and their colleagues. Our point-by-point responses are mentioned below in red.

We agree with the editorial colleague's suggestion to delete Fig 1a (the picture of the proband). It is now being excluded and the identifier (such as age) is also excluded from the main text.

We respectfully disagree with the change of Prune Belly Syndrome for Eagle-Barrett syndrome or Triad Syndrome.

Our co-senior author, Linda Baker, MD, is a 22-year, NIH-funded, physician surgeon scientist with molecular biology training and an actively practicing, board-certified pediatric urologist that care for over 50 children with prune belly syndrome in her clinical practice. Due to the rarity of this syndrome, Dr. Baker is nationally and internationally recognized as an expert in Prune Belly Syndrome. Dr. Baker dedicates a significant part of her clinical-academic career to understanding the causes and to improving the management of Prune Belly Syndrome, been the only NIH-funded scientist with an active R01 to study Prune Belly Syndrome.

She has 5 papers published (see below) in the area, where the words "Prune Belly Syndrome" is the way to call and describe the disease.

1. Genetic basis of **prune belly syndrome**: screening for HNF1 β gene. Granberg CF, Harrison SM, Dajusta D, Zhang S, Hajarnis S, Igarashi P, **Baker LA**. J Urol. 2012 Jan;187(1):272-8. doi: 10.1016/j.juro.2011.09.036.
2. Phenotypic severity scoring system and categorisation for **prune belly syndrome**: application to a pilot cohort of 50 living patients. Wong DG, Arevalo MK, Passoni NM, Iqbal NS, Jascur T, Kern AJ, Sanchez EJ, Satyanarayan A, Gattineni J, **Baker LA**.BJU Int. 2019 Jan;123(1):130-139. doi: 10.1111/bju.14524. Epub 2018 Sep 19.PMID: 30113772
3. Copy number variations in a population with **prune belly syndrome**. Iqbal NS, Jascur TA, Harrison S, Chen C, Arevalo MK, Wong D, Sanchez E, Grimsby G, Wilson K, **Baker LA**. Am J Med Genet A. 2018 Nov;176(11):2276-2283. doi: 10.1002/ajmg.a.40476. Epub 2018 Oct 4.PMID: 30285310

4. **Prune belly syndrome** in surviving males can be caused by Hemizygous missense mutations in the X-linked Filamin A gene. Iqbal NS, Jascur TA, Harrison SM, Edwards AB, Smith LT, Choi ES, Arevalo MK, Chen C, Zhang S, Kern AJ, Scheuerle AE, Sanchez EJ, Xing C, **Baker LA**. BMC Med Genet. 2020 Feb 21;21(1):38. doi: 10.1186/s12881-020-0973-x.PMID: 32085749

5. Modern management of and update on **prune belly syndrome**. Lopes RI, **Baker LA**, Dénes FT.J Pediatr Urol. 2021 Aug;17(4):548-554. doi: 10.1016/j.jpurol.2021.04.010. Epub 2021 Apr 24.PMID: 34016542

Additionally, a quick search on PubMed reveals 489 papers having “Prune Belly Syndrome” in the title as the way to name the disease. In contrast, only 11 papers have “Eagle-Barrett Syndrome” in the title, with 3 of these 11 also use “Prune Belly Syndrome” in the title. In the same way, a search on google reveal that even if you type “Eagle-Barrett Syndrome” the definition extract from National Organization for Rare disease (<https://rarediseases.org>) is “*Prune-Belly syndrome, also known as Eagle-Barrett syndrome, is a rare disorder characterized by partial or complete absence of the stomach (abdominal) muscles, failure of both testes to descend into the scrotum (bilateral cryptorchidism), and/or urinary tract malformations.*”

Further, an important USA patient support group named “Prune Belly Syndrome network (PBSN)”, a nonprofit organization, staffed by volunteers who serve as an advocate and additional line of support for those who have been born with prune belly syndrome, referrer to the disease as Prune Belly Syndrome.

Therefore, we believe that changing the Prune Belly Syndrome for Eagle-Barrett syndrome will decrease the diffusion of the paper and impact the search mechanism for the ones that are interesting to understand Prune Belly Syndrome.

For all these reasons, we would like to maintain Prune Belly Syndrome in the title and in the entire publication as the way to name the disease. We are happy to include Eagle-Barrett Syndrome as an additional name when we defined the disease, as well include Eagle-Barrett Syndrome as key word to facilitate the search mechanism.

We have the consent to deposit whole exome sequencing data and will be adding this information with final submissions.

Reviewer #1 (Remarks to the Author):

Overall, the edits and changes to the manuscript are good but there are still some things that require addressing.

1. “In review of the literature, a study published by Fotiou et al, 2015 identified one female patient with Generalized lymphatic dysplasia and additional clinical features including ‘prune’ belly.

This is indeed very supportive of the current findings!

Thank you.

2. “While there are gain-of-function and loss-of-function PIEZO1 mutations reported in the literature, to the best of our knowledge dominant negative PIEZO1 mutations have not been reported yet.”

A quick search reveals several articles that discuss and indeed show the possibility of dominant negative PIEZO1 variants (PMID: 32251670 PMID: 34489534 PMID: 34867393).

We agreed in the previous version that our data did not support dominant negative effect and these words are not mentioned in the paper. We think that it is best not to include the above-mentioned reference because while there is a possibility, it is not conclusively established in vivo. Furthermore, we do not use the term dominant negative which has a specific meaning in genetics.

3. “This proband harbors compound heterozygous PIEZO1 mutations, two inherited from the mother and one inherited from the father. These variants have not been tested in vitro or in vivo for functional significance.”

At least one of these variants has been tested in vitro PMID: 34489534.

We apologize for missing out on this point. We have now added the reference and revised our wordings on line 273

“One of these variants L939M has been tested functionally to compare with WT Piezo1 activity³⁹. The peak currents in response to stretch-activation were comparable between WT Piezo1 and L939M, however, modest rightward shift in the pressure-response curves were seen for the mutant³⁹. Additionally, of these three variants, the L939M and F2458L are reported as “uncertain significance” and R2456C is not reported at all in ClinVar”

4. “No increase in open probability (before and after pressure) is observed when both the mutants were expressed together (in same DNA quantities).”

I still find it hard to understand why higher pressures are not used to assess NPo over a range of pressures. This would strongly support the idea that these channels have reduced sensitivity to mechanical force and given the point below I think it becomes even more important.

Our focus was on obtaining high quality single channel data from each experiment for a longer duration to have enough events for histograms construction and obtaining single channel currents and open probability. We performed experiments at various pressure ranges such as at -10, -20, -30 and -50 mm Hg. The open probability acquired at various pressures is now included in supplementary Figure3 and the text associated with the figure is mentioned on lines 186-190.

“Additionally single channel open probability was also assayed in response to a range of pressures (from -10 to -50 mm Hg). In line with the data presented at -30 mm Hg, both the mutants S260R and S2211L exhibit lower NPo than WT Piezo1 at various applied pressure, strengthening the point that the mutants have lower sensitivity to pressure even at different pressures (Supplementary Figure 3).”

5. “We would like to show macro currents but consistent with our loss of function data, the mutants do not exhibit macro-currents in cell attached patches, as we can observe in WT Piezo1.”

This sentence for mutants that are trafficked equally as well as WT is worrying. This should be clearly noted in the text as a reason for not showing macro-currents and Figure S4 which shows similar labelling of Myc-tagged mutants to WT, which is now an even more key experiment, should be included in Figure 3 (preferably with some form of quantification). It's even more difficult to understand why so few channels are encountered given the influence of Yoda on these mutants. If they have a right shift in their pressure response curve and express at identical levels to WT it's hard to understand why macro-currents can't be recorded unless they have additional effects on inactivation. This should be discussed in the manuscript.

Our Myc-tag labelling to Piezo1 is to assess surface expression qualitatively. While it looks like mutants do make it to the surface of plasma membrane and there is no major trafficking problem, we could not perform quantification (on a few cells imaged).

We added this statement on line 208 in the results section.

“Further quantitative experiments with thousands of cells, such as flow cytometry, could be used to assess the exact percentage of mutant's surface expression in comparison to WT Piezo1, since our qualitative analysis imaged 9 cells per condition.”

It seems like with our experiment and transfection protocols and the size of the pipette used, we do not obtain macroscopic currents in cell-attached electrophysiology experiments even in the presence of yoda-1. We are unable to increase the number of channels in a patch, at best we have 4-6 channels which are not enough to perform reliable macroscopic analysis, which are usually in a Nano Ampere (nA) or 100s of Pico Ampere (pA) scale.

Minor:

Some of the original errors I spotted have not been corrected!! - Figure 4 still says uM & Line 845 mm Hg

Figure 4 μM is fixed now.

Line 166 assays

Changed now to assayed.

Line 276 there are data linking PIEZO1 to aortic aneurysm (PMID: 35082286) and structural abnormalities of the aortic valve (PMID: 32251670).

We have revised the wordings on line 286 and added the reference.

“Although PIEZO1 has not been directly connected with SVT and TAAD, in mice this protein serves as a key cardiovascular mechanotransducer, maintaining normal cardiovascular function⁴⁰⁻⁴⁴”

Line 219 punctuation

Thank you, it is fixed now.

Line 260 hydrops fetalis is superfluous.

Fixed now, we have removed the lymphedema because hydrops fetalis is clinically relevant term here.

Line 400 “and behave as compound heterozygous mutations,” doesn’t make sense. (Also line 215)

We have corrected the sentence now.

Line 861 n 9.

Thank you, the typo is corrected now.